# Super-enhancer-driven ZFP36L1 promotes PD-L1 expression in infiltrative gastric cancer

Xujin Wei[1,2†], Jie Liu[2†], Jia Cheng[3,4], Wangyu Cai[3,4], Wen Xie[3,4], Kang Wang[3,4], Lingyun Lin[3,4], Jingjing Hou[3,4], Jianchun Cai[2,3,4]*, Huiqin Zhuo[3,4]*

[1]Endoscopic Center, The First Affiliated Hospital, Fujian Medical University, Fuzhou, China; [2]The Graduate School of Fujian Medical University, Fuzhou, China; [3]Department of Gastrointestinal Surgery, Zhongshan Hospital of Xiamen University, Institute of Gastrointestinal Oncology, School of Medicine, Xiamen University, Xiamen, China; [4]Xiamen Municipal Key Laboratory of Gastrointestinal Oncology, Xiamen, China

**\*For correspondence:**
caijianchun@xmu.edu.cn (JC);
zhuohuiqin@xmu.edu.cn (HZ)

[†]These authors contributed equally to this work

**Competing interest:** The authors declare that no competing interests exist.

**Abstract** Gastric cancer (GC) is a major cause of cancer-related mortality worldwide. Despite the widespread recognition of tumor immunotherapy in treating unresectable GC, challenges, including ineffective immunotherapy and drug resistance, persist. Therefore, understanding the regulatory mechanisms of PD-L1, particularly in the context of super-enhancers (SEs) and zinc finger protein 36 ring finger protein-like 1 (ZFP36L1) RNA-binding protein, is crucial. In this study, we performed H3K27ac Cleavage Under Targets and Tagmentation (CUT&Tag) sequencing, investigated the heterogeneity of SEs between two GC subtypes with differential growth patterns, and revealed the immune escape signatures driven by ZFP36L1-SE in infiltrative GC through SEs inhibitors treatment. The regulation of ZFP36L1 to PD-L1 was evaluated by quantitative PCR, western blot, flow cytometry, and immunohistochemistry. Furthermore, we explored its regulatory mechanisms using a combination of molecular biology techniques, including luciferase reporter assay, GST/RNA pull-down, chromatin immunoprecipitation (ChIP)/RIP experiments, and in vivo functional assays. We demonstrated that ZFP36L1, driven by an SE, enhances IFN-γ-induced PD-L1 expression, with SPI1 identified as the specific transcription factor binding to ZFP36L1-SE. Mechanistically, ZFP36L1 binds to the adenylate uridylate-rich element in the 3′ untranslated region (3′UTR) of *HDAC3* mRNA, exacerbating its mRNA decay, and thereby facilitating PD-L1 abnormal transcriptional activation. Collectively, our findings provide mechanistic insights into the role of the SPI1-ZFP36L1-HDAC3-PD-L1 signaling axis in orchestrating immune escape mechanisms in GC, thereby offering valuable insights into the potential targets for immune checkpoint therapy in GC management.

## eLife assessment

The authors provide **useful** data to support the existence of a regulatory pathway starting with SPI1-driven ZFP36L1 expression, that goes on to downregulate HDAC3 expression at the transcript level, leading to PD-L1 upregulation due to implied enhanced acetylation of its promoter region. This is therefore an interesting pathway that adds to our understanding of how PD-L1 expression is controlled in gastric cancer. However, this is likely one of many possible pathways that impact PD-L1 expression, and the data are currently **incomplete** to support the claims made.

## Introduction

Gastric cancer (GC) remains the third leading cause of cancer-related mortality in China, thereby posing significant socio-economic challenges (*Qi et al., 2023*). The Ming classification, which categorizes GC into infiltrative and expanding subtypes based on distinct growth patterns, is pivotal in understanding the biological heterogeneity of this disease (*Ming, 1977*). Tumor immunotherapy, particularly targeting programmed cell death 1 ligand 1 (PD-L1; CD274), is a promising approach for unresectable GC (*Guan et al., 2023*). Notably, PD-L1, expressed on tumor cell surfaces, interacts with PD-1 receptors on cytotoxic T cells, leading to T-cell apoptosis and facilitating cancer immune evasion. Clinical studies, such as KEYNOTE-811, have demonstrated the efficacy of PD-L1 inhibitors, including pembrolizumab, especially in HER2-positive gastric adenocarcinoma study (*Janjigian et al., 2023*). Therefore, current guidelines recommend the use of checkpoint inhibitor therapy in GC cases with high PD-L1 combined positive scores. Despite these advancements, elucidating the regulatory mechanisms of PD-L1 is imperative owing to challenges associated with ineffective immunotherapy and drug resistance.

Super-enhancers (SEs), initially proposed by Richard A Young, are densely clustered transcriptionally active enhancers (*Whyte et al., 2013*). SEs exhibit distinct characteristics, including heightened enrichment of histone H3 acetylation at lysine 27 (H3K27ac), recruitment of numerous transcription factors (TFs) such as bromodomain-containing protein 4 (BRD4), MED1, and P300, and robust stimulation of target genes. Utilizing the rank ordering of super-enhancers (ROSE) algorithm, researchers can identify and explore cancer-associated SEs through techniques such as chromatin immunoprecipitation (ChIP)-seq and Cleavage Under Targets and Tagmentation (CUT&Tag) (*Liu et al., 2022*). In the epigenetic landscape of tumors, acquired SEs are implicated in promoting transcriptional dysregulation, contributing to oncogenesis, invasion, metabolic alterations, drug resistance, and the establishment of an immunosuppressive microenvironment (*Berico et al., 2023*; *Ye et al., 2023*). Previous studies have elucidated the unique SE landscape in mesenchymal-type GC and highlighted the role of EMT-related kinase NUAK1 at both transcriptional and epigenetic levels (*Ho et al., 2023*). However, further investigations are warranted to elucidate the effects of SEs during GC metastasis and immune evasion.

The zinc-finger protein 36 (ZFP36) gene family encodes three RNA-binding proteins: ZFP36, zinc finger protein 36 ring finger protein-like 1 (ZFP36L1), and ZFP36L2, characterized by two CCCH-type tandem zinc finger domains. These proteins recognize adenylate uridylate-rich elements (AREs) in the 3′ untranslated region (UTR) of target mRNAs, leading to mRNA deadenylation and degradation, ultimately diminishing protein synthesis. ZFP36 targets multiple inflammatory factors thereby playing a crucial role as an anti-inflammatory factor closely associated with inflammatory diseases such as psoriasis and arthritis (*Snyder and Blackshear, 2022*). *ZFP36L2* has been identified as an SE-activated gene with proto-oncogenic functions in GC (*Xing et al., 2019*). *ZFP36L1*, also known as *TIS11B/BRF1*, is primarily located in the q24.1 region of human chromosome 14, with a length of 8599 bases long. The gene exhibits three annotated transcripts, with NM_004926.4 being the predominant transcript. The encoded protein comprises 338 amino acids and contains two tandem zinc finger structural domains, a nuclear localization signal, and a nuclear export signal region. It is distributed across the nucleus, cytoplasm, and subcellular structural apparatus. Despite ongoing debates regarding its role, ZFP36L1 remains a subject of interest in tumor biology.

We previously identified differential protein expression patterns in expanding and infiltrative GC, notably observing increased levels of MED1 and P300 in infiltrative GC (*Figure 1—figure supplement 1A*; *Hong et al., 2023*). Based on these findings, we postulated that infiltrative GC may exhibit heightened enhancer activity and dysregulated transcription. To validate this hypothesis, we aimed to compare the distribution of SEs between Ming-expanding and infiltrative GC. ZFP36L1-SE was then identified as the key SE in infiltrative GC. Additionally, we aimed to elucidate the significance of SE-driven ZFP36L1 in tumor immune evasion and the SPI1-ZFP36L1-HDAC3-PD-L1 signaling axis. Collectively, we believe that our findings would contribute to a deeper understanding of immune checkpoint therapies for GC.

## Results

### SE heterogeneity between two subtypes of GC with differential growth patterns

Six GC samples were collected, and H3K27ac CUT&Tag sequencing was performed for the first time to identify of GC SEs (*Figure 1A*). The two GC growth patterns did not significantly differ in terms of typical enhancers (*Figure 1—figure supplement 1B*). However, a bimodal H3K27ac enrichment was observed in infiltrative GC compared with that in expanding GC (*Figure 1B*). A total of 1057 and 819 infiltrative and expanding SE peaks, respectively, were obtained using the ROSE algorithm. Over 50% of these peaks were located in the non-coding regions such as exons and introns, and their predicted target genes were transcribed to produce non-coding RNAs; the peaks distributed in transcription start and termination sites activated the promoters and directly drove the transcription of protein-coding genes (*Figure 1C*). Collectively, these data targeted 240 infiltrative and 173 expanding SE-driven protein-coding genes (*Figure 1D*, *Figure 1—figure supplement 1C*).

Gene Ontology (GO) and Kyoto Encyclopedia of Genes and Genomes (KEGG) analyses displayed that cellular response to epidermal growth factor stimulus, ERK1 and ERK2 cascade, and MAPK signaling pathway signatures were enriched in both expanding and infiltrative SE-driven genes; cadherin binding, positive regulation of protein localization to cell periphery, and regulation of GTPase activity were enriched in expanding SE-driven genes; negative regulation of T-cell proliferation, response to tumor necrosis factor, regulation of epithelial cell migration, and 3′UTR-mediated mRNA destabilization pathway signatures were enriched in infiltrative SE-driven genes (*Figure 1E*). After the overall survival analysis for GC patients in The Cancer Genome Atlas (TCGA) datasets, we screened 16 of these infiltrative SE-driven genes with prognostic value (*Figure 1—figure supplement 1D*). Unsupervised hierarchical clustering revealed that infiltrative SE-driven clusters showed significant infiltration of memory, regulatory, and helper T cells (*Figure 1F–H*). The protein expression present in 70 PD-L1-positive GC tumor tissues was assessed, and high immunohistochemical (IHC) scores were determined in PD-L1 infiltrative GC compared with those in expanding GC (*Figure 1I*). These results helped describe the SE-driven immune escape signatures of infiltrative GC.

### *ZFP36L1* as an SE-driven oncogene in infiltrative GC

*ZFP36L1* was selected among these 16 genes based on Friends analysis and a comprehensive exploration of the TCGA data (*Figure 2A*). The mRNA expression in GC correlated positively with high T stage, tumor grade, diffuse type, and *Helicobacter pylori* infection (*Figure 2B*) and negatively correlated with overall survival, disease-specific survival, and progression-free interval (*Figure 2—figure supplement 1*). Notably, high *ZFP36L1* expression represented high tumor immune dysfunction and exclusion (TIDE) scores, T-cell infiltration and high *CD274* (also known as PD-L1) expression (*Figure 2C–E*). Therefore, we speculated that ZFP36L1 is a key molecule for immune escape in infiltrative GC, and ZFP36L1-SE (chr14:68806839–68816867) is a probable cause of transcriptional dysregulation as it is situated upstream of the *ZFP36L1* promoter (*Figure 2F and G*). This genomic region harbors two typical enhancers (E1:68806839–68807740 and E2:68816088–68816867) and shows a plethora of histone acetylation enrichments in infiltrative GC tissues, as well as in MKN45 and AGS cell lines. A similar trend was observed at the protein level in this study. In the 12 paired infiltrative GC samples, ZFP36L1 protein expression was higher in 10 primary neoplasms than that in normal adjacent tissues (*Figure 2H*). Additionally, various GC cell lines exhibited high ZFP36L1 expression compared with that in the normal gastric epithelial cell line, GES-1 (*Figure 2I*).

To validate whether ZFP36L1 was driven by the SEs, XGC-1, and MKN45 cells were treated with the SE inhibitors THZ1 and JQ1. We observed that THZ1 and JQ1 inhibited mRNA and protein expression of ZFP36L1 in a concentration- and time-dependent manner, respectively (*Figure 2J*). Moreover, ChIP experiments revealed that JQ1 decreased H3K27ac enrichment in ZFP36L1-SE region, especially in E1 region (*Figure 2K*). These findings suggest that *ZFP36L1* is a key SE-driven oncogene involved in infiltrating GC.

### ZFP36L1 promotes IFN-γ-induced PD-L1 expression

Given the aforementioned results, we speculated that ZFP36L1 contributes to heightened PD-L1 expression in infiltrative GC. Typically, IFN-γ derived from T cells triggers PD-L1 overexpression on

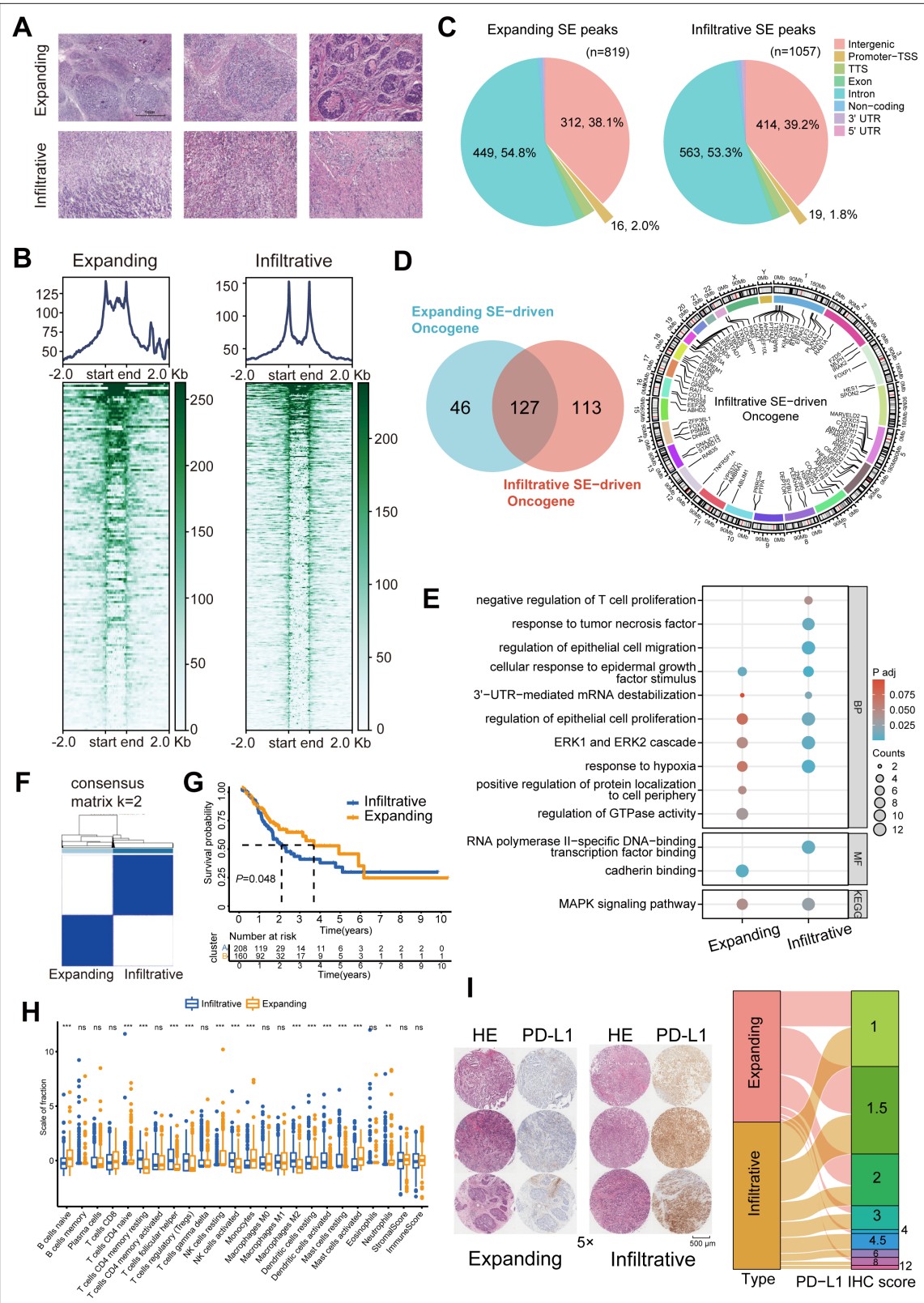

**Figure 1.** Immune escape signatures of Ming infiltrative gastric cancer (GC) driven by super-enhancers (SEs). (**A**) Hematoxylin-eosin staining of GC. (**B**) SE peaks of H3K27ac histone modifications. (**C**) Distribution of H3K27ac SE peaks. (**D**) Venn diagrams of SE-driven protein-coding genes and chromosomal landscape of infiltrative SE-driven genes. (**E**) Gene Ontology-Kyoto Encyclopedia of Genes and Genomes (GO-KEGG) pathway enrichment for SE-driven genes. (**F**) Unsupervised hierarchical clustering using 16 prognostic genes in GC patients from The Cancer Genome

*Figure 1 continued on next page*

*Figure 1 continued*

Atlas (TCGA) datasets. (**G**) Kaplan-Meier survival curves of two subgroups. (**H**) Immune infiltration analysis. (**I**) Immunohistochemical (IHC) scores of programmed death-ligand 1 (PD-L1) in 70 GC tissues. TSS, transcription start site; TTS, transcription termination site.

The online version of this article includes the following figure supplement(s) for figure 1:

**Figure supplement 1.** Epigenetic heterogeneity between expanding and infiltrative gastric cancer.

the surface of GC cells. In vitro experiments were conducted to simulate the immune microenvironment by adding exogenous IFN-γ at a concentration of 40 ng/mL, exacerbating PD-L1 expression on GC cells. However, in *ZFP36L1* knockdown XGC-1 and MKN45 GC cell lines, IFN-γ-induced PD-L1 transcription levels were lower compared with that in the control group (*Figure 3A*). Conversely, we observed a significant increase in IFN-γ-induced PD-L1 mRNA expression in XGC-2 and MGC803 cell lines overexpressing *ZFP36L1* (*Figure 3B*). These results indicate that ZFP36L1 potentiates IFN-γ-induced PD-L1 transcription.

At the protein level, knocking down *ZFP36L1* in the XGC-1 and MKN45 cell lines similarly reduced IFN-γ-induced PD-L1 protein expression (*Figure 3C*). Conversely, overexpression of *ZFP36L1* in the XGC-2 and MGC803 cell lines facilitated IFN-γ-induced PD-L1 protein expression (*Figure 3D*). Flow cytometry revealed a consistent trend in the expression levels of PD-L1 on the tumor cell membrane surface. Knockdown of *ZFP36L1* decreased the surface PD-L1 fluorescent signal in IFN-γ-induced XGC-1 and MKN45 cells compared with that in the control group (*Figure 3E*), whereas overexpression of ZFP36L1 demonstrated a higher fluorescent signal intensity in IFN-γ-induced XGC-2 and MGC803 cells (*Figure 3F*). These results validate the conclusion that ZFP36L1 expression is positively correlates with IFN-γ-induced PD-L1 expression at the protein level.

## SPI1 binding to the SE region of *ZFP36L1*

To identify the upstream TF driving the SE-associated oncogene, the MEME-ChIP online tool was used to identify TF-binding sites in ZFP36L1-SE. Given the high GC content of E2, which is JQ1-insensitive, only the sequence motifs enriched in E1 (motif ID: GRGGMAGGARG) were examined. *Figure 4A* lists the predicted TFs and corresponding DNA motifs, including ETS transcription factor (TF) family (such as SPI1, ELF1, and ETS1) and E2F TF family (such as E2F1 and E2F6). SPI1, ELF1, and E2F1 were confirmed using another online analysis tool based on ChIP-seq data from the Signaling Pathways Project (*Figure 4—figure supplement 1A*). Among these, SPI1 is a known interferon regulatory TF that modulates PD-L1 mRNA expression (*Li et al., 2022*). Moreover, querying TCGA data revealed an association between SPI1 expression and poor prognosis (*Figure 4B*). Consequently, we speculated that SPI1 is a tissue-specific TF driving ZFP36L1 transcription by activating the SE region.

To validate this hypothesis, MKN45 cells were transfected with *SPI1*, *ELF1*, *E2F1*, and control plasmids. The real-time PCR (RT-PCR) results indicated that only transfection with SPI1 upregulated the *ZFP36L1* mRNA level (*Figure 4C*). Similarly, *SPI1* overexpression increased the ZFP36L1 protein levels in MKN45 and MGC803 cells (*Figure 4D*). TCGA data revealed that *SPI1* mRNA expression positively correlated with PD-L1 expression in patients with stomach cancer (*Figure 4E*). Overexpression of *SPI1* increased IFN-γ-induced PD-L1 protein amount and fluorescent signal at membrane surfaces in GC cells, but concurrent knockdown of *ZFP36L1* could reverse the results of PD-L1 expression (*Figure 4F* and *Figure 4—figure supplement 1B*). These findings suggest that SPI1 regulates PD-L1 expression in a ZFP36L1-dependent manner.

Further analysis of protein-protein interaction networks using STRING indicated that SPI1 might interact with BRD4 and P300 proteins, which are SE-labeled molecules (*Figure 4G*). Subsequently, the formation of the transcriptional complex comprising SPI1 and BRD4 was investigated. Exogenous BRD4 co-immunoprecipitated with exogenous SPI1 in 293T cells co-transfected with two plasmids, as observed using the anti-Flag affinity resin (*Figure 4H*). In MGC803 cells, endogenous SPI1 immunoprecipitated with endogenous BRD4 (*Figure 4I*). Additionally, we constructed a prokaryotic expression system of SPI1 and BRD4, and performed protein purification and GST pull-down assays (*Figure 4J*). These results indicate that SPI1 and BRD4 directly bind in vitro, reflecting the specificity of SE-associated TFs.

To assess SE activity and TF occupancy, the E1 fragment was inserted into the luciferase reporter pGL4-Basic vector. In 293T cells, luciferase signals were enhanced when the *SPI1* plasmid, instead of

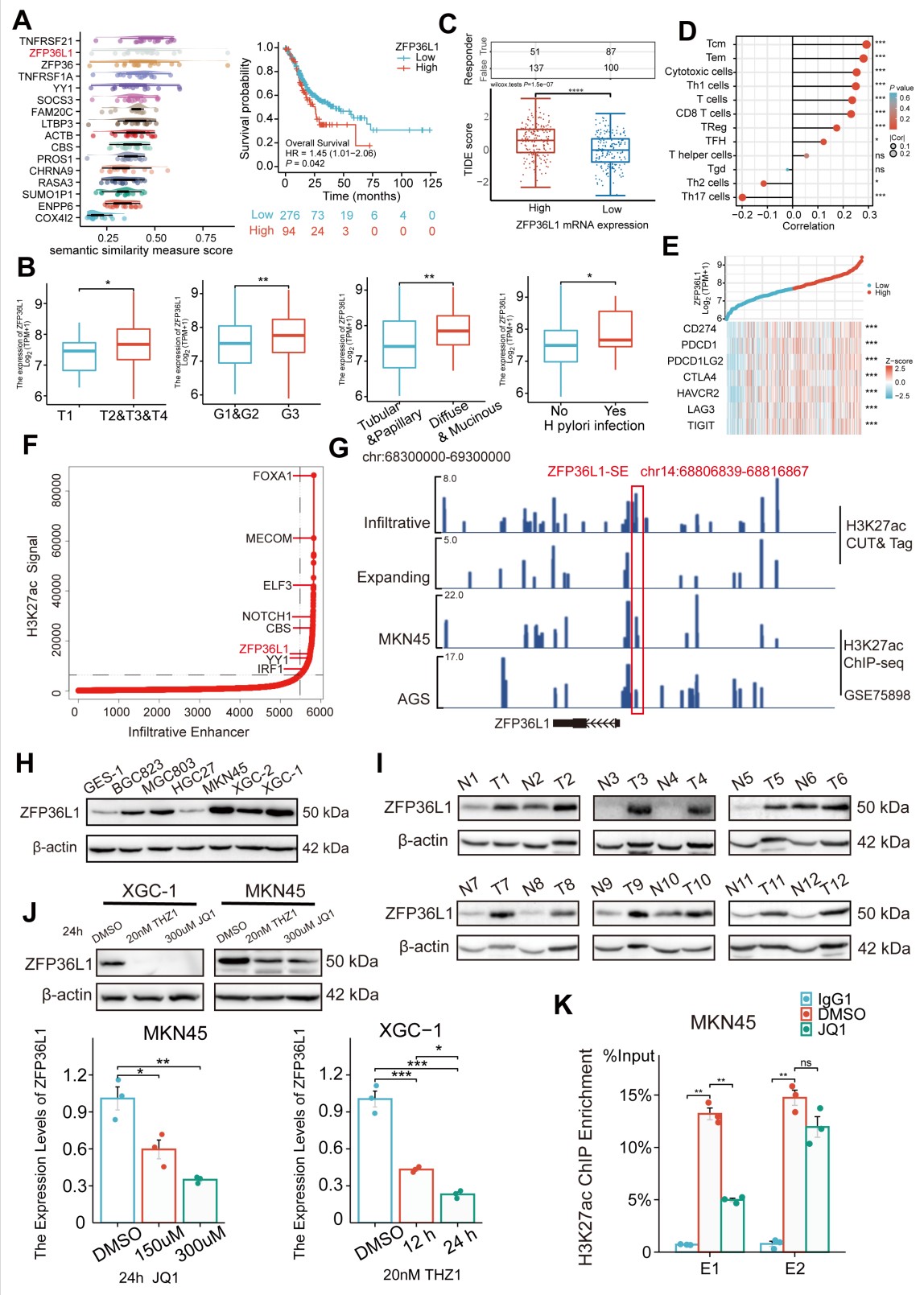

**Figure 2.** Expression levels of *ZFP36L1* in infiltrative gastric cancer (GC) driven by ZFP36L1-SE. (**A**) Friends analysis of 16 super-enhancer (SE)-driven prognostic genes. (**B**) Correlations between clinical characteristics and the *ZFP36L1* mRNA expression in The Cancer Genome Atlas (TCGA). (**C**) Tumor immune dysfunction and exclusion scores in high and low expression levels of *ZFP36L1* groups. (**D**) Correlation between immune infiltration cells and the mRNA expression level of *ZFP36L1* in TCGA. (**E**) Correlation between *ZFP36L1* mRNA expression and immune checkpoints in TCGA. (**F**) H3K27ac

*Figure 2 continued on next page*

*Figure 2 continued*

signals of SEs and target genes in infiltrative GC. (**G**) H3K27ac signals of ZFP36L1-SE in GC. Protein expression of *ZFP36L1* in (**H**) 6 GC cell lines and (**I**) 12 tumor and paired adjacent normal tissues of patients with infiltrative GC. (**J**) Expression level of *ZFP36L1* after SE inhibition treatment (n=3). (**K**) H3K27ac signals of ZFP36L1-SE after SE inhibition treatment (n=3). ***, p<0.001; **, p<0.01; *, p<0.05; ns, p≥0.05. (**B–C**) t-Test, (**D–E**) *Spearman's* correlation, (**J**) one-way ANOVA with post hoc Tukey HSD test, and (**K**) Welch's ANOVA with a Games-Howell post hoc test were used for statistical analysis.

The online version of this article includes the following source data and figure supplement(s) for figure 2:

**Source data 1.** PDF file containing original western blots for *Figure 2H–J*.

**Source data 2.** Original files for western blot analysis displayed in *Figure 2H–J*.

**Figure supplement 1.** Kaplan-Meier for (**A**) disease-specific survival and (**B**) progress-free interval plot of ZFP36L1.

---

*ELF1* or *E2F1*, was co-transfected with pGL4-E1 (*Figure 4K*). This indicates a well-defined binding site for SPI1 in the ZFP36L1-E1 region, consistent with previous RT-PCR results. The six predicted DNA sequence motifs in the E1 region are distributed among four DNA-binding sites, with sites C and D containing two adjacent DNA sequence motifs.

Subsequently, we performed ChIP experiments using SPI1 antibodies to immunoprecipitate bound DNA fragments in MGC803 cells, followed by RT-PCR analysis of the products. Among these endogenous SPI1-binding sites, the abundance of the site C product was the highest (*Figure 4L*). Consequently, four truncated E1 fragments containing different binding sites were inserted into the luciferase reporter vector (E1A-E1D). As anticipated, we observed significantly enhanced luciferase activity upon co-transfection with the SPI1 and pGL4-E1C plasmids (*Figure 4M*). Furthermore, two motifs totaling 15 bp were deleted from site C to construct a plasmid with a deletion mutation. The transcriptional activation of SPI1 was abrogated in the deletion mutation group compared with that in the wild-type group (*Figure 4N*). These results suggest that site C is the SPI1-binding region in ZFP36L1-SE.

## Upregulation of PD-L1 mediated by *HDAC3* mRNA decay

To identify downstream target genes, the mRNA transcripts bound to ZFP36L1 were searched against the RNAct website. A bibliometric analysis via PubMed suggested that 51 gene transcripts may be associated with PD-L1 expression (*Figure 5A*). Representative molecules were selected to perform preliminary verification, which revealed that *HDAC3* mRNA expression was significantly reduced in MGC803 cells overexpressing ZFP36L1 (*Figure 5B*). Conversely, protein levels of HDAC3 increased in XGC-1 and MKN45 cells with *ZFP36L1* knockdown (*Figure 5C*). HDAC3 reportedly represses PD-L1 transcription through histone deacetylation (*Wang et al., 2020*; *Yokoyama et al., 2021*). Likewise, we observed that overexpression of *HDAC3* does not directly affect PD-L1 promoter activity (*Figure 5D*), but inhibits PD-L1 transcription through histone H3K27 deacetylation in the promoter region (*Figure 5E and F*). Additionally, histone H3K27 deacetylation is also detrimental to the transcriptional activity of ZFP36L1-E1, thereby providing a negative feedback loop that inhibits ZFP36L1 expression (*Figure 5G and H*). Therefore, we conducted rescue experiments involving *HDAC3* plasmid co-transfection. The PD-L1-inducing effects of IFN-γ were reinforced in cell lines overexpressing *ZFP36L1* (*Figure 5I and J*). However, PD-L1 protein levels and surface fluorescent signals decreased when *HDAC3* was simultaneously overexpressed. Additionally, the effects of *SPI1* overexpression were rescued when *HDAC3* expression was concurrently restored in MGC803 and MKN45 cells (*Figure 5K and L*). These findings suggest that SE-driven ZFP36L1 positively regulates PD-L1 expression by inhibiting HDAC3. The inhibition of HDAC3 additionally provide positive feedback to promote *ZFP36L1* transcription.

Actinomycin D was employed to assess the rate of *HDAC3* mRNA degradation. *ZFP36L1* knockdown in MKN45 cells led to an increased half-life of *HDAC3* mRNA post-actinomycin D treatment, indicating enhanced mRNA stability and suppression of mRNA decay. Conversely, *ZFP36L1* overexpression in MGC803 cells promoted mRNA degradation, impairing *HDAC3* mRNA stability (*Figure 5M*). This observation suggests that ZFP36L1 influences *HDAC3* mRNA decay.

To further elucidate the mechanism of RNA-binding protein leading to mRNA deadenylation, we conducted a series of experiments. Initially, we validated the direct binding of ZFP36L1 protein to the 3'UTR region of *HDAC3* mRNA via RNA immunoprecipitation. Owing to the lack of highly potent ZFP36L1 antibodies, RNA fragments were enriched using an anti-FLAG antibody in MGC803 cells overexpressing exogenous FLAG-tagged ZFP36L1. While we observed no significant change in

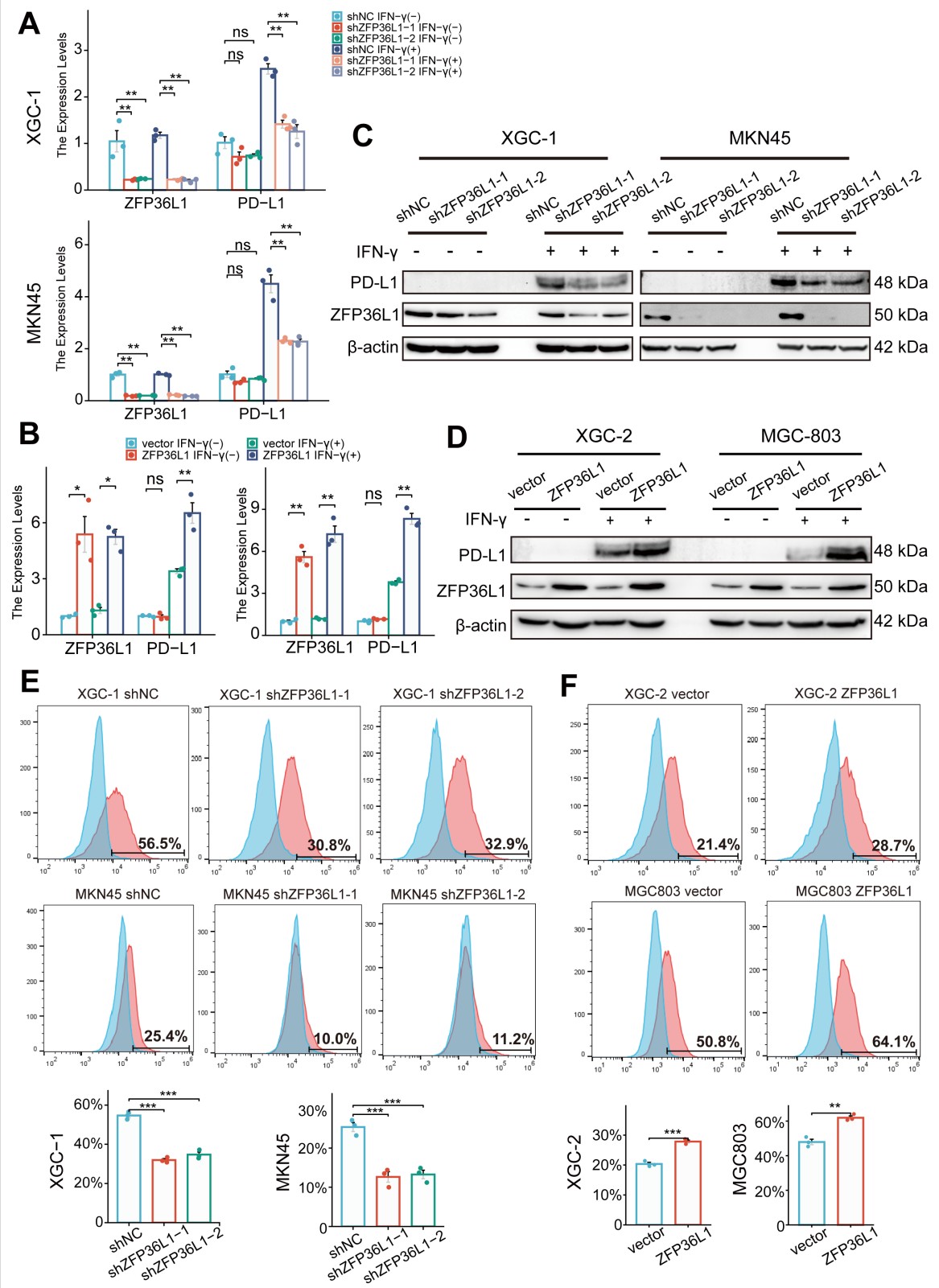

**Figure 3.** ZFP36L1 promotes IFN-γ-induced PD-L1 expression. (**A**) mRNA and (**C**) protein expression of PD-L1 in gastric cancer (GC) cell lines with or without *ZFP36L1* knockdown (n=3). (**B**) mRNA and (**D**) protein expression of PD-L1 in GC cell lines with or without *ZFP36L1* overexpression (n=3). Fluorescent signal of the PD-L1 membrane protein in GC cell lines with or without *ZFP36L1* (**E**) knockdown and (**F**) overexpression (n=3). ***, p<0.001;

*Figure 3 continued on next page*

*Figure 3 continued*

**, p<0.01; *, p<0.05; ns, p≥0.05. (**A–B**) Welch's ANOVA with a Games-Howell post hoc test, (**E**) one-way ANOVA post hoc Tukey HSD, and (**F**) t-test were used for statistical analysis.

The online version of this article includes the following source data for figure 3:

**Source data 1.** PDF file containing original western blots for *Figure 3*.

**Source data 2.** Original files for western blot analysis displayed in *Figure 3*.

HPRT1 mRNA (negative control), the abundance of HDAC3 mRNA fragments was amplified in the immunoprecipitated products (*Figure 5N*).

Subsequently, the binding site in the ARE of the 3′UTR was identified. Classical binding motifs of the ZFP36 protein family, such as 'WTTTW' and 'WWTTTWW,' were considered. The 3′UTR sequences of *HDAC3* mRNA were examined, revealing that only 'ATTTA' fulfilled the requirements. The 3′UTR sequences and pmirGLO vectors were used to introduce a point mutation into the luciferase reporter gene plasmid, where the central base sequence of 'ATTTA' was substituted with 'ACCCA'. ZFP36L1 impaired the transcriptional activity of the wild-type plasmid but not the mutant plasmid, directly demonstrating the regulatory site of mRNA decay-promoting activity (*Figure 5O*).

Finally, the CCCH-type zinc finger domains were examined to confirm their indispensable role in RNA binding. Accordingly, we mutated two cysteine (C) residues at positions 135 and 173 to arginine (R) (*Figure 5P*). *HDAC3* mRNA probes were synthesized and co-incubated with extracts of MGC803 cells transfected with a mutant (C135R-C173R) or wild-type plasmid. RNA pull-down assay results revealed that the mutant ZFP36L1 did not bind to the 3′UTR of HDAC3 mRNA. Collectively, our findings suggest that ZFP36L1 potentiated PD-L1 expression by promoting *HDAC3* mRNA decay.

## Correlation between ZFP36L1 and PD-L1 in vivo

Protein expression levels were assessed in 70 PD-L1-positive GC tumor tissues via IHC staining. The IHC score of ZFP36L1 positively correlated with PD-L1 and SPI1 while exhibiting a negative correlation with HDAC3 (*Figure 6A–C*). These results substantiate the upstream-downstream relationship observed in human tissue specimens.

Wild-type mice with normal immune function were chosen to explore these regulatory relationships in vivo. The mouse GC cell line MFC was substituted with the MC38 cell line owing to its poor tumorigenic capacity in wild-type mice. Subsequently, stable knockdown mouse *Zfp36l1* cell lines were established for subcutaneous tumor formation and tail vein assay of lung metastasis (*Figure 6D*). Although *Zfp36l1* knockdown did not impact the size of subcutaneous tumors (*Figure 6E*), it led to reduced mRNA and protein expression of *Pdl1* (*Figure 6F and G*). Furthermore, the mice in the *Zfp36l1* knockdown group exhibited fewer metastatic pulmonary nodules compared with that in mice in the control group (*Figure 6H*), with representative lung tissue sections demonstrating strong positivity for Cd8α (*Figure 6I*). Conversely, the number of pulmonary metastatic nodules was not decreased in the T-cell-deficient nude mice injected with *Zfp36l1* knockdown cells (*Figure 6J*). These findings suggest potential immunotherapeutic benefits associated with targeting ZFP36L1. *Figure 7* illustrates the SPI1-ZFP36L1-HDAC3-PD-L1 signaling axis, depicting the interconnected regulatory relationship among these molecules.

## Discussion

In the present study, we identified the SE heterogeneity between two subtypes of GC with differential growth patterns. SEs have emerged as a whole novel branch within tumor epigenetics. The direct experimental validation of SEs involves assessing the expression of target genes after the deletion of SE regions via the CRISPR/Cas9 system. In the context of bladder cancer, FOSL1 directly binds to SNHG15-SE, stimulating the Wnt signaling pathway through interactions with SNHG15-CTNNB1, thereby driving malignant behavior (*Tan et al., 2023*). Exploring different loci enables the identification of crucial SE peaks and their functional roles. For instance, *TOX2* acts as an SE-driven oncogene in extranodal natural killer/T-cell lymphoma, with its transcription governed by the SE-specific TFs, RUNX3. Targeting SE regions with various single-guide RNAs has confirmed the pathogenicity of TOX2-SE (*Zhou et al., 2023*). While the CRISPR/Cas9 system was not directly investigated in this

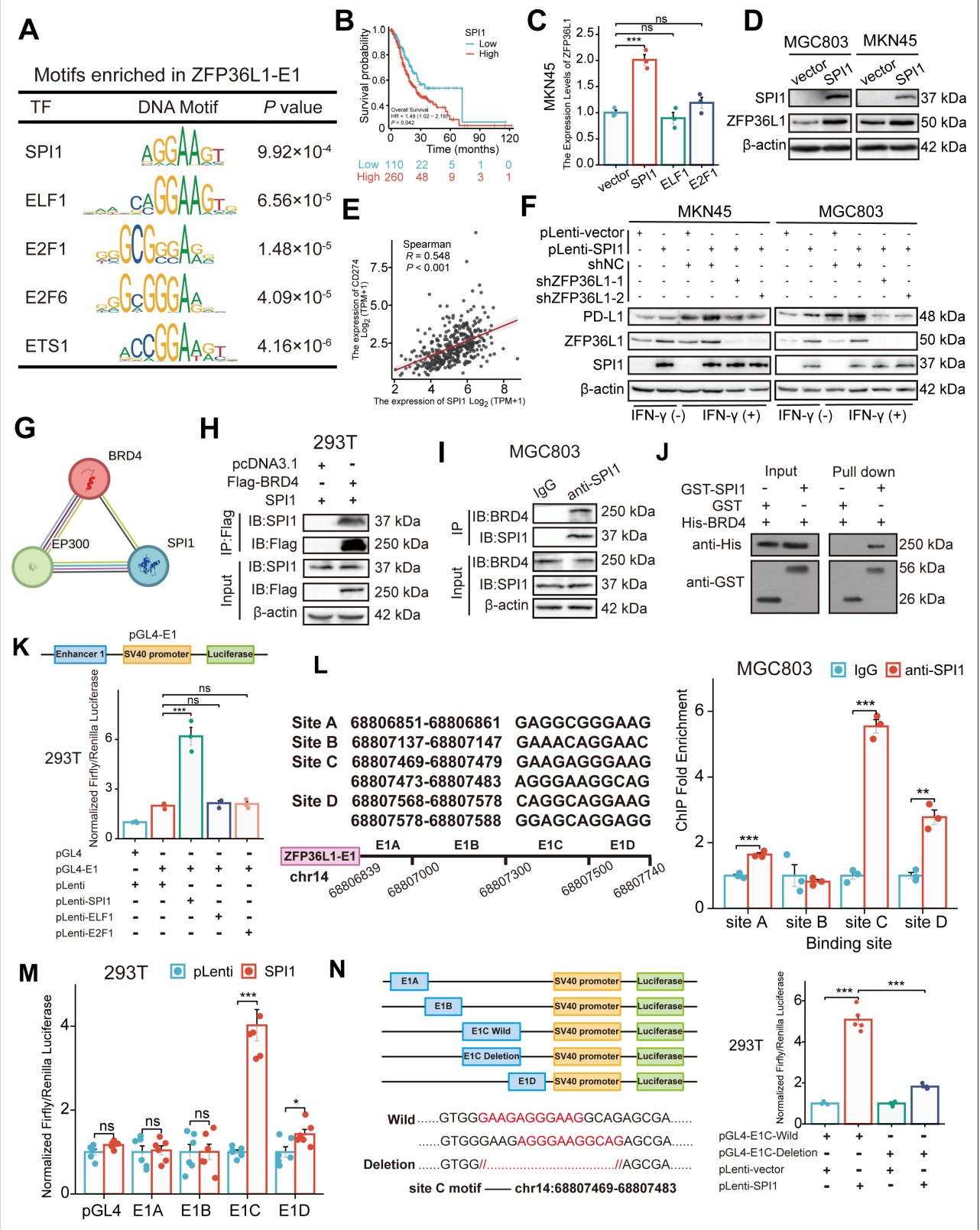

**Figure 4.** SPI1 binds to the ZFP36L1-SE region and drives the regulation of PD-L1. (**A**) Schematic of transcription factor (TF) motif enrichment in ZFP36L1-E1. (**B**) Kaplan-Meier survival plot of SPI1 in The Cancer Genome Atlas (TCGA). (**C**) The mRNA expression of ZFP36L1 after TFs plasmid transfection (n=3). (**D**) The ZFP36L1 protein expression in cell lines overexpressing *SPI1*. (**E**) Correlation between SPI1 and PD-L1 mRNA expression in TCGA. (**F**) PD-L1 protein expression in simultaneous *SPI1* overexpression and *ZFP36L1* knockdown cells. (**G**) Prediction of SPI1-BRD4-P300 binding on

*Figure 4 continued on next page*

*Figure 4 continued*

the STRING website. Co-immunoprecipitation between (**H**) exogenous SPI1 and BRD4 in 293T cells, or (**I**) endogenous SPI1 and BRD4 in MGC803. (**J**) SPI1 directly interacts with BRD4 in vitro by GST pull-down experiment. (**K**) ZFP36L1-E1 binding of different TFs detected using dual-luciferase assay (n=3). (**L**) SPI1 enriched regions in ZFP36L1-E1 detected by chromatin immunoprecipitation (ChIP) assay (n=3). (**M**) Different binding sites of SPI1 in ZFP36L1-E1 detected using dual-luciferase assay (n=6). (**N**) Wild-type and motif-deletion mutant E1C binding of SPI1 detected using dual-luciferase (n=5). \*\*\*, p<0.001; \*\*, p<0.01; \*, p<0.05; ns, p≥0.05. (**C** ,**K**) One-way ANOVA post hoc Tukey HSD test, (**L**) t-test, (**M**) Welch's t-test, and (**N**) Welch's ANOVA with a Games-Howell post hoc test were used for statistical analysis.

The online version of this article includes the following source data and figure supplement(s) for figure 4:

**Source data 1.** PDF file containing original western blots for *Figure 4*.

**Source data 2.** Original files for western blot analysis displayed in *Figure 4*.

**Figure supplement 1.** Transcriptional regulation of PD-L1 by SPI1.

study, the biological mechanism underlying ZFP36L1-SE was indirectly corroborated using ChIP, BRD4 co-immunoprecipitation, and luciferase assays. The interactions between promoters and SEs are reportedly mediated by CCCTC-binding factor loops. Recently, researchers have identified novel regulatory elements within SEs, termed 'facilitators,' which highlights the significance of SE integrity (*Blayney et al., 2023*). Furthermore, SEs can transcribe SE RNA, which is beneficial for chromatin openness (*Li et al., 2023b*). Collectively, further investigation into the functional architecture of SEs is warranted.

Understanding the mechanisms governing SE formation is currently a prominent area of research. *PDZK1IP1* is an acquired SE-driven oncogene in primary colorectal tumors, with its encoded protein regulating the pentose phosphate pathway to promote tumor redox capability. Although PDZK1IP1-SE does not initially exist in colon cancer cell lines, its production is stimulated by abundant inflammatory factors within the tumor microenvironment (*Zhou et al., 2022*). These acquired somatic mutations exert influence over transcriptional epigenetic regulation. Acquired SEs in GC exhibit a high binding affinity for CDX2 or HNF4α, alongside enriched chromatin interactions and cancer-associated single-nucleotide polymorphisms (SNPs) (*Ooi et al., 2016*). Liu et al. identified a genetic variation (SNP rs10470352) within the SOX2-SE region through a joint analysis of genome-wide association studies and HiChIP. This sequence variation enables TP73/RUNX3 occupancy, promoting active chromatin and upregulating *SOX2* expression (*Liu et al., 2023b*). However, the infiltration-specific nature of ZFP36L1-SE raises questions regarding whether its pro-tumor function is attributable to somatic mutations.

We demonstrate for the first time that ZFP36L1 driven by ZFP36L1-SE promotes IFN-γ-induced PD-L1 expression. Given that the mRNA of numerous inflammatory factors and oncogenes harbor AREs in their 3′UTR, ZFP36 family proteins are typically regarded as anti-tumor and anti-inflammatory agents. Loh et al. proposed that *ZFP36L1* deletion mutations are characteristic of urological tumors owing to their targeting of hypoxic mRNAs or cell cycle markers, such as *HIF1A*, *CCND1*, and *E2F1*, thereby promoting their degradation (*Loh et al., 2020*). The authors also reported the inhibition of the HDAC family by ZFP36L1. In small-cell lung cancer, inhibition of lysine-specific demethylase 1 triggers *ZFP36L1* expression, leading to the decay of *SOX2* mRNA and attenuation of tumor neuroendocrine differentiation (*Chen et al., 2022a*). This highlights the complex role of ZFP36L1, thereby cautioning about simple dichotomy.

Histone deacetylase 3 (HDAC3), a key member of the HDAC family (which includes HDAC1 through HDAC11), is pivotal in regulating histone acetylation and deacetylation (*Haga-Yamanaka et al., 2024*). By modulating chromatin accessibility, HDAC3 influences transcriptional processes. It often forms co-repressor complexes with other chromatin remodeling factors to inhibit histone acetylation (*Zhang et al., 2023*). Beyond its role in histone modification, HDAC3 has been implicated in the regulation of non-histone proteins, broadening its functional scope. Notably, HDAC3 antagonizes the activity of P300, catalyzing the deacetylation of core histones (H2A, H2B, H3, and H4) and contributing to transcriptional repression. Recent studies highlight its collaboration with HDAC1 in catalyzing histone de-succinylation at promoter regions—an epigenetic mark associated with transcriptional activation (*Li et al., 2023a*).

Novel treatment regimen in combination with anti-PD-L1 mainly enhanced CD8+ T-cell infiltration (*Zhao et al., 2024*). Several studies have proposed that reducing PD-L1 expression enhances the tumor-killing effect of cytotoxic T lymphocytes in vitro and reduces primary tumor foci in vivo.

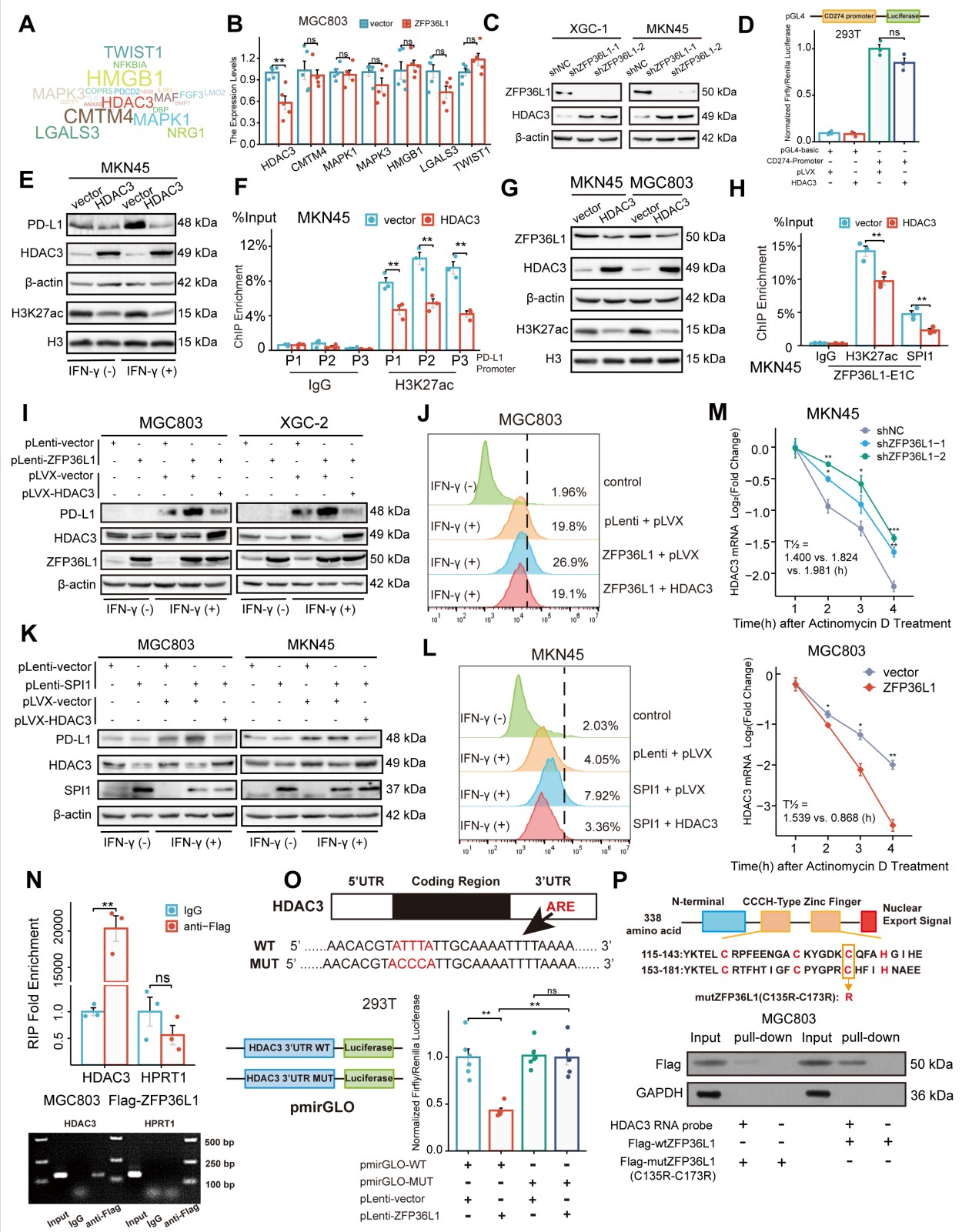

**Figure 5.** ZFP36L1 positively regulates PD-L1 by activating *HDAC3* mRNA decay. (**A**) Word cloud of predicted ZFP36L1 target genes. (**B**) The mRNA expression of predicted target genes in MGC803 cell overexpressing ZFP36L1 (n=5). (**C**) HDAC3 protein expression in *ZFP36L1* knockdown cells. (**D**) Effect of HDAC3 on *CD274* promoter activity in 293T using dual-luciferase assay (n=3). (**E**) Correlation between changes of histone H3K27 acetylation and PD-L1 protein expression in MKN45 cells overexpressing *HDAC3*. (**F**) Chromatin immunoprecipitation (ChIP) assay showing the histone H3K27

*Figure 5 continued on next page*

*Figure 5 continued*

acetylation levels of *CD274* promoter regions in MKN45 cells overexpressing HDAC3 (n=3). (**G**) Correlation between changes of histone H3K27 acetylation and ZFP36L1 protein expression. (**H**) The H3K27ac and SPI1 enrichment of ZFP36L1-E1C regions using ChIP assay (n=3). (**I, J**) PD-L1 protein expression in simultaneous *ZFP36L1* and *HDAC3* overexpression cells. (**M**) *HDAC3* mRNA decay in *ZFP36L1* knockdown and overexpression cells after actinomycin D treatment. (**N**) ZFP36L1 mRNA-binding level by RNA-binding protein immunoprecipitation (n=3). (**O**) ZFP36L1 mRNA-binding site in AU-rich element (ARE) of 3′UTR confirmed using dual-luciferase assay (n=6). (**P**) CCCH-type zinc finger domain of ZFP36L1 protein binding to *HDAC3* mRNA confirmed using RNA pull-down assay. ARE, adenylate uridylate- (AU-) rich element; 3′UTR, 3′ untranslated region. ***, p<0.001; **, p<0.01; *, p<0.05; ns, p≥0.05. (**B, F**) Wilcoxon rank sum test, (**D**) one-way ANOVA post hoc Tukey HSD test, (**H**) t-test, (**N**) Welch's t-test, and (**O**) Kruskal-Wallis with Dunn's test were used for statistical analysis.

The online version of this article includes the following source data for figure 5:

**Source data 1.** PDF file containing original western blots for *Figure 5*.

**Source data 2.** Original files for western blot analysis displayed in *Figure 5*.

Conversely, findings from this study suggest that PD-L1 expression is associated with immune evasion in metastatic foci (***Klement et al., 2023***). The upstream regulation of PD-L1 expression occurs at both mRNA and protein levels. The well-established JAK-STAT/IRF1 signaling pathway primarily governs IFN-γ-induced PD-L1 transcription, with certain oncogenes such as *DENR*, *UBR5*, and *SOX10* promoting tumor immune evasion via activation of this pathway (***Yokoyama et al., 2021***; ***Chen et al., 2022b***; ***Wu et al., 2022***). Additionally, CD274-associated transcriptional regulators include *HIF1A*, *MYC*, and *YBX1* (***Ru et al., 2024***). Shang et al. demonstrated that MTHFD2 facilitates *O*-GlcNAc glycosylation of the c-MYC protein during metabolic reprogramming, thereby activating PD-L1 transcription (***Shang et al., 2021***). However, ZFP36L1 disrupts *HDAC3* mRNA, leading to PD-L1-mediated immune evasion independent of the aforementioned TFs. It destabilizes the regulatory region of *CD274*, enhances histone acetylation, and promotes chromatin accessibility. Likewise, depletion of *ZNF652* disrupts the formation of the HDAC1/2-MTA3 co-repressor complex, resulting in elevated PD-L1 expression in triple-negative breast cancer (***Liu et al., 2023a***). Conversely, the m6A reader IGF2BP1 binds to the 3′UTR, initiating m6A modification and stabilizing PD-L1 mRNA (***Jiang et al., 2024***).

The regulatory mechanisms governing PD-L1 predominantly involve post-translational modifications and subcellular relocalization. K48-linked deubiquitination by USP2 and N-glycosylation by B4GALT1 stabilize PD-L1 and prevent its degradation. Both K48-linked de-ubiquitination by USP2 and N-glycosylation by B4GALT1 are able to stabilize and prevent degradation of PD-L1 protein (***Cui et al., 2023***; ***Kuang et al., 2023***). TRIM28 overexpression in GC reportedly prevents proteasomal degradation of PD-L1 protein through SUMOylation, as well as promotes PD-L1 transcription by activating the mTOR and TBK1-IRF1 pathways (***Ma et al., 2023***). However, ZFP36L1 primarily exerts a negative influence on signaling pathways and is not associated with protein modification. In terms of subcellular relocalization, the mitochondrial membrane protein ATAD3A in paclitaxel-resistant breast cancer patients obstructs the mitochondrial translocation and autophagic degradation of PD-L1, resulting in increased membranous PD-L1 expression (***Xie et al., 2023***). ZFP36L1 reportedly forms a subcellular compartment with the endoplasmic reticulum (ER), facilitating the formation of membrane-less organelles in the TIS11B (ZFP36L1) granule ER domain. Within this domain, the interaction between SET and CD47/CD274, which harbor multiple AREs, promotes membrane protein expression (***Ma and Mayr, 2018***); follow-up studies are warranted to comprehensively elucidate its role.

In conclusion, our findings elucidate the regulation of the SPI1-ZFP36L1-HDAC3-PD-L1 axis in infiltrative GC and highlight immune escape signatures driven by SEs. These findings offer complementary insights for the application of immunotherapy in GC.

## Materials and methods
### Cell lines and tissues

MKN45 and MGC803 are derived from human GC; 293T is from human embryonic kidney cells; and MC38 is sourced from mouse colorectal cancer cells. All cell lines were acquired from the Institute of Cell Biology (Shanghai, China). Additionally, infiltrative GC cell line XGC-1 (China patent No. CN103396994A) and expanding GC cell line XGC-2 (China patent No. CN103387963B) were constructed by our team and have been authenticated (***Xu et al., 2016***; ***Peng et al., 2020***). The cell

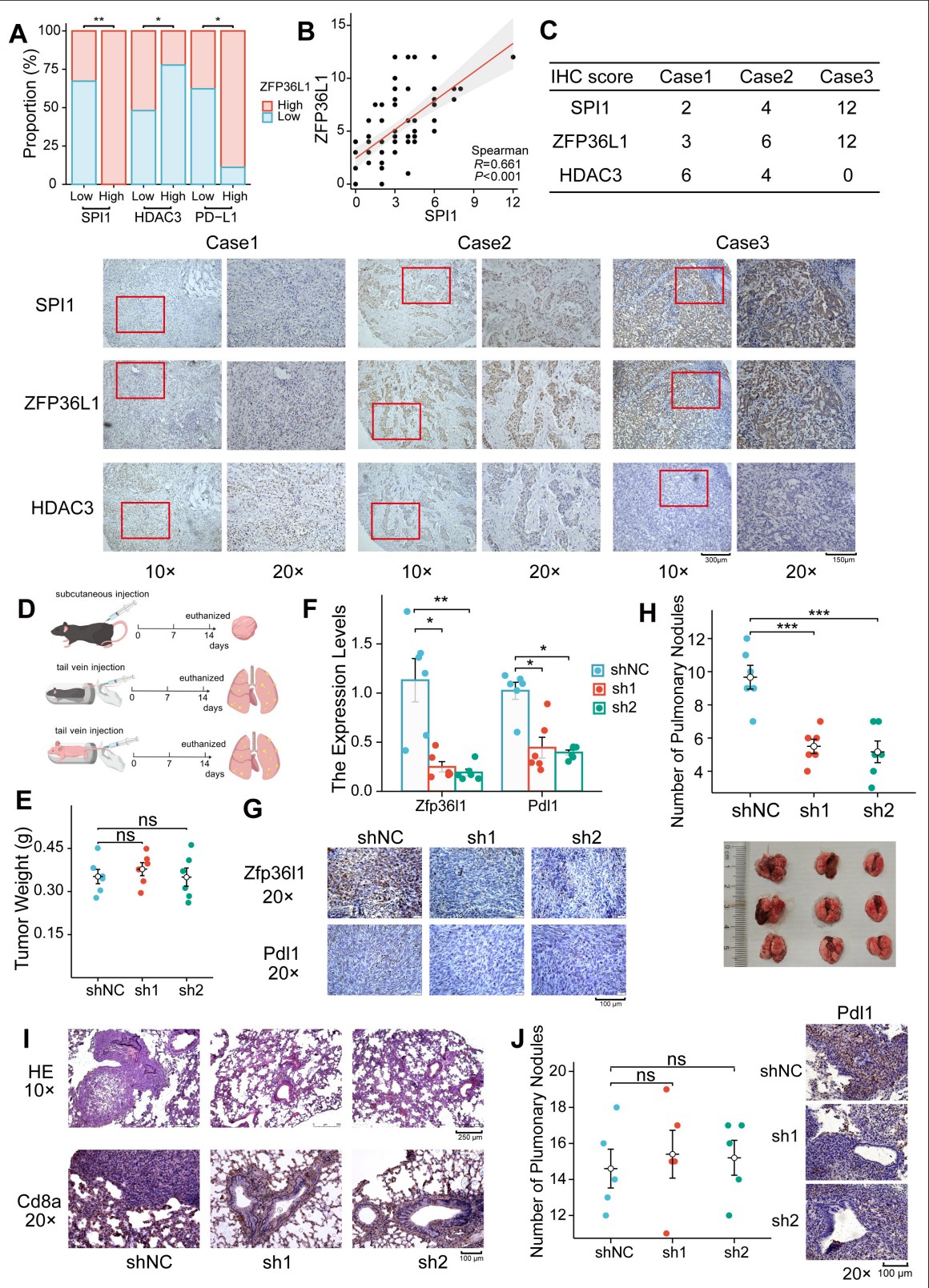

**Figure 6.** Positive correlation between ZFP36L1 and PD-L1 in vivo. (**A**) Immunohistochemical (IHC) staining scores of SPI1, ZFP36L1, and HDAC3 protein in 70 PD-L1-positive gastric cancer (GC) tumor tissues. (**B**) Scatter plot to compute correlation between SPI1 and ZFP36L1. (**C**) Three patients with infiltrative GC shown as representative images. (**D**) Schematic diagram of experiments in C57BL/6J and BALB/c-nu mice. (**E**) Subcutaneous tumor weight from mice injected with control and ZFP36L1 knockdown cells (n=6). (**F**) The PD-L1 mRNA expression in subcutaneous tumors (n=6). (**G**) IHC

*Figure 6 continued on next page*

*Figure 6 continued*

staining of ZFP36L1 and PD-L1 protein in subcutaneous tumors. (**H**) Number of pulmonary metastases in different groups of C57BL/6J mice after tail vein injection of control and *ZFP36L1* knockdown cells (n=6). (**I**) Hematoxylin & eosin (HE) and IHC staining of CD8α in pulmonary metastases. (**J**) Number of pulmonary metastases in different groups of BALB/c-nu mice after tail vein injection of control and *ZFP36L1* knockdown cells (n=5). ***, p<0.001; **, p<0.01; *, p<0.05; ns, p≥0.05. (**E, H, J**) One-way ANOVA post hoc Tukey HSD test and (**F**) Kruskal-Wallis with Dunn's test were used for statistical analysis.

lines were confirmed to be correct using Short Tandem Repeat profiling by American Type Culture Collection at the time of purchase, and they are regularly tested for mycoplasma contamination, with results consistently negative. 293T and MC38 were grown in high-glucose DMEM with 10% FBS, and other cell lines were cultured in RPM1640 medium. Six GC tissues for H3K27ac CUT&Tag sequencing were obtained from patients undergoing resection of primary GC at the Zhongshan Hospital of Xiamen University. GC tissue microarray (HStmA180Su19) containing 70 PD-L1-positive cancer tissues was acquired from Outdo Biotech (Shanghai, China).

## Statistical methods and bioinformatic analysis

If the variable is numerical and the sample size is ≤5000, a normality test will be conducted. For numerical variables that satisfy the normal distribution and pass the variance chi-squared test, two-group comparisons will be performed using the t-test, and three-group comparisons will use one-way ANOVA. If the data satisfy the normal distribution but fail the variance chi-squared test, two-group

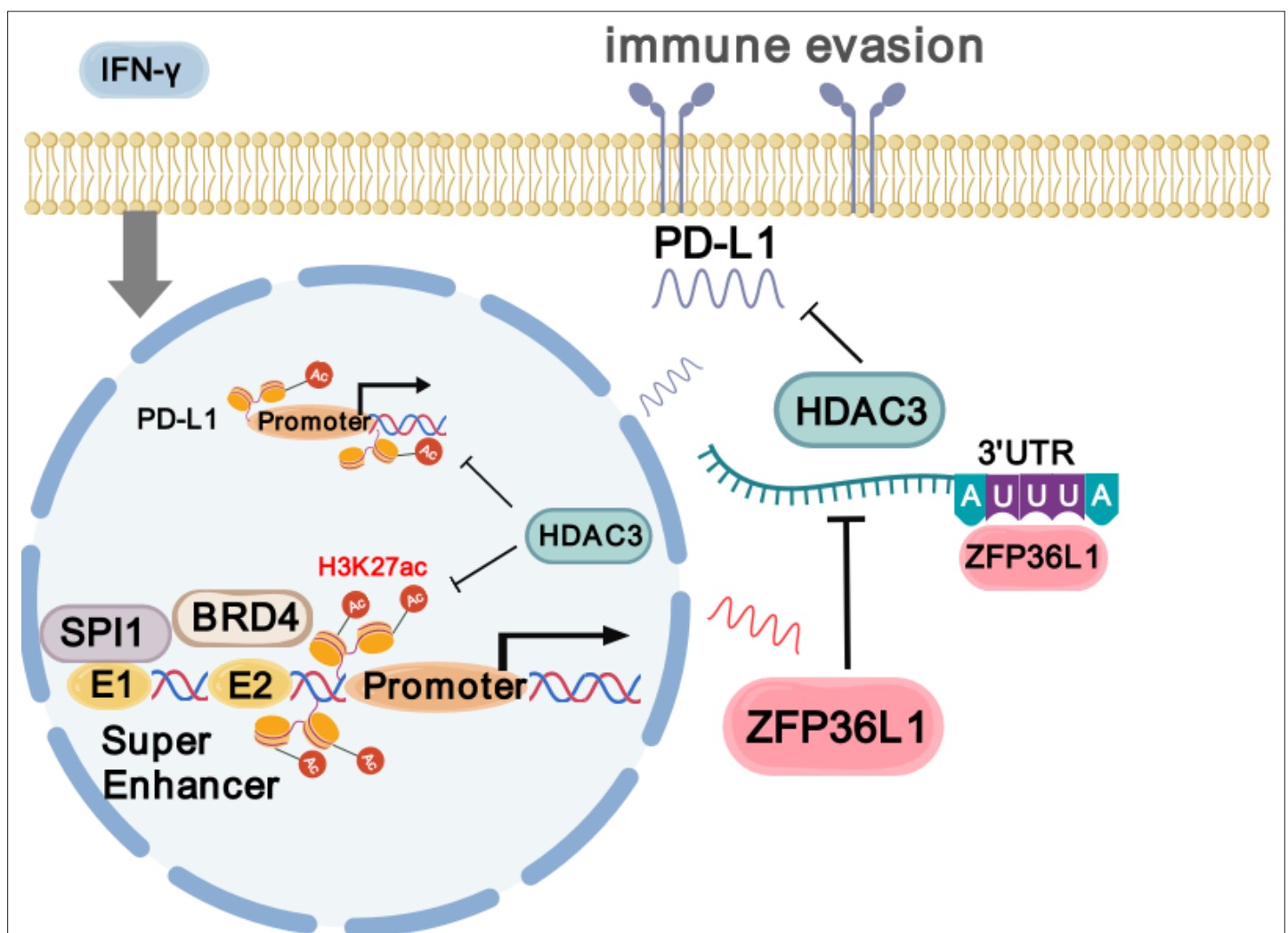

**Figure 7.** Schematic diagram of SPI1-ZFP36L1-HDAC3-PD-L1 signaling axis (created with gdp.renlab.cn).

comparisons will be conducted using Welch's t-test, and three-group comparisons will use Welch's one-way ANOVA. If the data do not satisfy the normal distribution assumption, two-group comparisons will use Welch's one-way ANOVA. For normally distributed data, two-group comparisons will use the Wilcoxon test, and three-group comparisons will use the Kruskal-Wallis test.

Bioinformatics analyses and graphing of experimental results were performed using R (4.2.1) version, involving the following R packages: ggplot2 [3.3.6] car [3.1-0], and stats [4.2.1] for histograms, GOSemSim [2.22.0] for Friends analysis, VennDiagram [1.7.3], clusterProfiler [4.4.4] for GO-KEGG analysis, survival [3.3.1] and survminer for Kaplan-Meier survival curves, ConsensusClusterPlus for unsupervised clustering, ggalluvial [0.12.3] for Sankey diagram, circlize [0.4.15] for the localization maps, ggwordcloud [0.6.0] for word cloud. The ROSE algorithm: http://younglab.wi.mit.edu/super_enhancer_code.html. MEME-ChIP: https://meme-suite.org/meme/tools/meme (*Bailey et al., 2015*). Primer design was performed using Primer Premier 5. Mechanism mapping: http://gdp.renlab.cn.

## CUT&Tag and ChIP

Epi CUT and Tag Kit (Epibiotek, Guangzhou, China) was used to perform the CUT&Tag experiment with five steps: mixing the cell suspension with magnetic beads (10 µL ConA Beads and $1×10^5$ cells), antibody binding (297 µL Wash Buffer, 3 µL 5% Digitonin, 12 µL 25× pAb Mix, and 3 µL antibody), incubation with Pa-Tn5 transposome, labeling (50 µL Tagmentation Buffer for 1 hr at room temperature), library construction, and sequencing. ROSE algorithm was programmed in Python (v3.9) to identify SEs. The data was visualized using IGV 2.14.1.

Sonication ChIP Kit (ABclonal, Wuhan, China) was used to perform the ChIP experiment with five steps: crosslinking (1% formaldehyde solution 10 min, 10× Glycine Solution 5 min), cell nuclear extraction ($1×10^7$ cells, centrifugation at 5000×*g* for 5 min at 4°C), ultrasonication (25% power with 3 s on/off periods for a total of 8 min, ideal fragment size: 200–500 bp), immunoprecipitation, eluting the chromatin, de-crosslinking, DNA purification, and qPCR. The primary H3K27ac antibody (ABclonal, Cat. No. A7253) or SPI1 antibody (Abcam, Cat. No. ab227835) was used at 10 µg per sample. Primer sequences are available in *Supplementary file 1a*.

## Western blot

Antibodies used for the western blot assay were as follows: ZFP36L1 (Abcam, Cat. No. ab230507); HDAC3 (Abcam, Cat. No. ab76295); CD274 (Proteintech, Cat. No. 66248-1-lg); HA-Tag (Proteintech, Cat. No. 51064-2-AP); DDDDK-Tag (ABclonal, Cat. No. AE005); β-actin (ABclonal, Cat. No. AC026).

## HE and IHC staining

Fresh tissue samples are fixed in formalin overnight, then dehydrated using an automatic dehydration instrument, embedded in paraffin, and stored as tissue blocks. Sections of 4 µm thickness are prepared and dried. For deparaffinization, slides are baked at 65°C and sequentially immersed in decreasing concentrations of ethanol and water. HE staining involves hematoxylin staining, decolorization in hydrochloric acid ethanol, and eosin staining. Antigen retrieval is performed using high-pressure steam, followed by immunostaining with specific antibodies and DAB color development. Finally, slides are counterstained with hematoxylin, dehydrated, and mounted. IHC scoring is based on the extent and intensity of positive tumor cell staining, with scores ≥5 indicating high expression and <5 indicating low expression.

## Real-time PCR

The RT-PCR experiment involves three main steps. First, total RNA is extracted using a cell total RNA extraction kit, where cells are lysed, and RNA is purified through a series of centrifugation and washing steps, followed by elution with RNase-free water. Second, RNA reverse transcription is performed using the HiFi-MMLV cDNA first-strand synthesis kit, where 2 µg of RNA is mixed with various reagents and incubated to synthesize cDNA. Finally, RT-PCR is conducted with diluted cDNA templates, using a 2× UltraSYBR Mixture and specific primers. The RT-PCR program includes initial denaturation, 40 amplification cycles, and a melting curve analysis to verify product specificity.

## RNA-binding protein immunoprecipitation

First, MGC803 cells transfected with the pcDNA3.1-ZFP36L1-Flag plasmid are harvested, lysed in RIP lysis buffer, and stored at –80°C. Magnetic beads are prepared by washing with RIP wash buffer and then incubated with Flag antibody or IgG. The cell lysates are thawed, and a portion is set aside as a 5% input control before being added to the bead-antibody complex for overnight incubation at 4°C. After washing the beads, RNA is purified by resuspending the complexes in Proteinase K buffer, followed by phenol-chloroform extraction and ethanol precipitation. The resulting RNA is then reverse-transcribed into cDNA, and real-time quantitative PCR is performed to detect the products. Results are normalized against the input control to analyze fold differences.

## Co-immunoprecipitation

The co-immunoprecipitation experiment utilizes two methods: DYKDDDDK-G1 affinity resin and Protein A/G magnetic beads. For the affinity resin method, 50 µL of resin is pre-treated with TBS buffer, followed by incubation with 400 µL of prepared protein sample overnight at 4°C. The resin is washed and denatured with Loading Buffer, heated, and the supernatant is collected for western blot detection. In the magnetic bead method, 30 µL of beads are pre-treated with binding/washing buffer and then incubated with diluted antibody for 2 hr at 4°C. After washing, the antigen sample is added, and the mixture is incubated again. The beads are then washed, and denaturing elution is performed with Loading Buffer, followed by heating and supernatant collection for western blot analysis.

## GST pull-down

First, competent cells are transformed with pET-32a-His-BRD4, pGEX-4T-2-GST-SPI1, and empty vector plasmids, followed by induction with IPTG during the logarithmic growth phase for 24 hr at 18°C. After collecting and lysing the bacterial pellets, His-BRD4 is purified using Ni-NTA columns with gradient washing and elution in imidazole-containing PBS. For GST-SPI1, the cleared supernatant is incubated with GST beads, washed, and eluted with GST Elution Buffer. Finally, the GST pull-down assay is performed by equilibrating glutathione resin, binding the GST-fusion target protein, preparing the prey protein, and eluting the bait-prey complex for subsequent gel electrophoresis analysis.

## RNA pull-down

First, MGC803 cells are transfected with pcDNA3.1-ZFP36L1-Flag and pcDNA3.1-mutZFP36L1-Flag plasmids, followed by cell lysis after 48 hr. Next, single-stranded RNA probes containing the HDAC3 sequence are synthesized from a clone plasmid using T7 RNA polymerase, purified, and biotinylated overnight. The biotinylated RNA is recovered and dissolved in DEPC-treated water. Finally, the assay is conducted by pre-treating magnetic beads, binding the biotinylated RNA to streptavidin-coated beads, and performing RNA-protein binding, washing, and elution of the RNA-protein complex, followed by western blot analysis to evaluate the results.

## Dual-luciferase reporter assay

293T cells in 12-well plates were transfected with 0.8 µg *Renilla* luciferase plasmid, 0.8 µg firefly luciferase plasmid (vector: pGL4 or pmirGLO), and 0.8 µg TF plasmid. Dual-luciferase reporter assay kit (Vazyme, Nanjing, China) was used to detect the enzyme/substrate reactions, and the fluorescence values were normalized. Plasmids are available in *Supplementary file 1b*.

## Flow cytometry

$1 \times 10^6$ cells was washed and resuspended in 100 µL FACS buffer (98% PBS+2% FBS) after digestion. 1 µL fluorescently labeled primary CD274 antibody (ABclonal, Cat. No. A22305) was added per tube and incubated for 20 min at room temperature. Cells labeled for PD-L1 were detected by flow cytometry (excitation light: 647 nm; emission light: 664 nm) and the data was visualized using FlowJo (v10.8.1).

## Actinomycin D-induced mRNA decay

Cells were seeded in six-well plates and allowed to grow to 80%. Add actinomycin D to a final concentration of 1 µg/mL, and cellular RNA in each plate was collected every other hour. RNA extraction was performed using the FastPure Cell/Tissue Total RNA Isolation Kit (Vazyme, Nanjing, China), and

RT-PCR was performed using UltraSYBR Mixture (Cowin Biotech, Jiangsu, China). 18s rRNA was used as an internal reference gene, and half-lives were calculated from linear-log graphs.

## Animals and treatment

All the procedures involving animals were conducted in accordance with the ethical principles and were approved by the Institutional Animal Care and Use Committee and Laboratory Animal Management Ethics Committee at Xiamen University (number of animal experiments ethical approval: XMULAC20220268). 5-Week C57BL/6J or BALB/c mice were purchased from Xiamen University Laboratory Animal Center, and the center was responsible for the daily feeding. At the beginning of each experiment, all mice were randomly assigned to control or experimental groups each containing six mice. MC38 cells were infected with shZfp36l1 expression lentivirus, and $2\times10^6$ cells were injected subcutaneously or $1\times10^6$ cells were injected into the tail vein. Two weeks later, mice were sacrificed and dissected for the HE and IHC staining.

## Acknowledgements

We thank Xin Chen, Hao Zhang, Shihao Rao, and Cheng Huang for the experimental assistance.

## Additional information

### Funding

| Funder | Grant reference number | Author |
|---|---|---|
| National Natural Science Foundation of China | No. 81871979 | Jianchun Cai |
| National Natural Science Foundation of China | No. 82272894 | Huiqin Zhuo |
| Natural Science Foundation of Fujian Province | No. 2021J02056 | Jianchun Cai |
| Natural Science Foundation of Fujian Province | No. 2020CXB048 | Huiqin Zhuo |
| Natural Science Foundation of Fujian Province | 2021D026 | Huiqin Zhuo |
| Natural Science Foundation of Fujian Province | No. 2023J011594 | Jia Cheng |

The funders had no role in study design, data collection and interpretation, or the decision to submit the work for publication.

### Author contributions

Xujin Wei, Data curation, Formal analysis, Investigation, Visualization, Methodology, Writing - original draft; Jie Liu, Data curation, Software, Formal analysis, Validation, Investigation, Methodology, Writing – review and editing; Jia Cheng, Supervision, Funding acquisition, Methodology, Project administration, Writing – review and editing; Wangyu Cai, Formal analysis, Validation, Visualization, Methodology, Writing – review and editing; Wen Xie, Formal analysis, Investigation, Visualization, Methodology, Writing – review and editing; Kang Wang, Software, Supervision, Validation; Lingyun Lin, Validation, Investigation, Methodology, Writing – review and editing; Jingjing Hou, Software, Supervision, Validation, Visualization, Methodology, Writing – review and editing; Jianchun Cai, Conceptualization, Resources, Data curation, Software, Funding acquisition, Validation, Project administration, Writing – review and editing; Huiqin Zhuo, Conceptualization, Software, Supervision, Funding acquisition, Validation, Investigation, Project administration, Writing – review and editing

## Author ORCIDs

Xujin Wei (ID) https://orcid.org/0000-0002-1527-6400
Jie Liu (ID) https://orcid.org/0000-0003-3307-1975
Jia Cheng (ID) https://orcid.org/0000-0003-3116-4035
Jianchun Cai (ID) https://orcid.org/0000-0001-6217-6165
Huiqin Zhuo (ID) https://orcid.org/0000-0001-8322-4197

## Ethics

The study was conducted in accordance with the Declaration of Helsinki. This study obtained written informed consent from the participating subjects, who agreed to the secondary research use of their tissue samples. The Ethics Committee of Zhongshan Hospital of Xiamen University reviewed and approved this study: xmzsyyky-2020-126, 3 September, 2020.

All the procedures involving animals were conducted in accordance with the ethical principles and were approved by the Institutional Animal Care and Use Committee and Laboratory Animal Management Ethics Committee at Xiamen University (Number of animal experiments ethical approval: XMULAC20220268).

Reviewer #1 (Public Review): https://doi.org/10.7554/eLife.96445.2.sa1
Reviewer #2 (Public Review) https://doi.org/10.7554/eLife.96445.2.sa2
Author response https://doi.org/10.7554/eLife.96445.2.sa3

---

# Additional files

## Supplementary files

• Supplementary file 1. Primer sequences and plasmids. (A) Primer sequences and (B) plasmids used in this study.

• MDAR checklist

## Data availability

H3K27ac CUT& Tag datasets performed in this study are deposited at the NCBI Gene Expression Omnibus under the accession GSE275349. TCGA data were downloaded from the UCSC Xena website http://xena.ucsc.edu/ ; RNA-binding protein prediction website: http://rnact.crg.eu ; ChIP-seq prediction of transcription factors: https://www.signalingpathways.org.

The following dataset was generated:

| Author(s) | Year | Dataset title | Dataset URL | Database and Identifier |
|---|---|---|---|---|
| Wei X, Liu J | 2024 | Super-enhancer H3K27ac CUT& Tag sequencing of six human gastric cancer tissue samples | https://www.ncbi.nlm.nih.gov/geo/query/acc.cgi?acc=GSE275349 | NCBI Gene Expression Omnibus, GSE275349 |

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
