## [Editor Report · eLife assessment]

The authors provide **useful** data to support the existence of a regulatory pathway starting with SPI1-driven ZFP36L1 expression, that goes on to downregulate HDAC3 expression at the transcript level, leading to PD-L1 upregulation due to implied enhanced acetylation of its promoter region. This is therefore an interesting pathway that adds to our understanding of how PD-L1 expression is controlled in gastric cancer. However, this is likely one of many possible pathways that impact PD-L1 expression, and the data are currently **incomplete** to support the claims made.

---

## [Referee Report · Reviewer #1 (Public Review)]

In this paper, the authors provide data to support the existence of a regulatory pathway starting with SPI1-driven ZFP36L1 expression, that goes on to downregulate HDAC3 expression at the transcript level, leading to PD-L1 upregulation due to implied enhanced acetylation of its promoter region. This is therefore an interesting pathway that adds to our understanding of how PD-L1 expression is controlled in gastric cancer. However, this is likely one of many possible pathways that impact PD-L1 expression, which is likely equally important. Thus, while potentially interesting, this is more additive information to the literature rather than a fundamentally new concept/finding.

Overall, there are many experiments presented, which appear to be of good quality, however, there are a number of issues with this that need attention. Moreover, the text is often difficult to follow, partly due to the standard of English, but mainly due to the sparsity of detail in the results section and figure legends. Thus providing an overall assessment of data conclusiveness is not possible at this time. This is exacerbated by frequently extrapolating conclusions beyond what is actually shown in an individual experiment.

Major issues:

(1) All the figure legends need to expand significantly, so it is clear what is being presented. All experiments showing data quantification need the numbers of independent biological replicates to be added, plus an indication of what the P-values are associated with the asterisks (and the tests used).

(2) Related to point 1, the description of the data in the text needs to expand significantly, so the figure panels are interpretable. Examples are given below but this is not an exhaustive list.

(3) The addition of "super-enhancer-driven" to the title is a distraction. This is the starting point but the finding is portrayed by the last part of the title. Moreover, it is not clear why this is a super enhancer rather than just a typical enhancer as only one seems to be relevant and functional. I suggest avoiding this term after initial characterisations.

(4) The descriptions of Figures 1B, C, and D are very poor. How for example do you go from nearly 2000 SE peaks to a couple of hundred target genes? What are the other 90% doing? What is the definition of a target gene? This whole start section needs a complete overhaul to make it understandable and this is important as is what leads us to ZFP36L1 in the first place.

(5) It is impossible to work out what Figures 1F, H, and I are from the accompanying text. The same applies to supplementary Figure S1D. Figure 1G is not described in the results.

(6) What is Figure 2A? There is no axis label or description.

(7) Why is CD274 discussed in the text from Figure 2E but none of the other genes? The rationale needs expanding.

(8) Figure 2G needs zooming in more over the putative SE region and the two enhancers labelling. This looks very strange at the moment and does not show typical peak shapes for histone acetylation at enhancers.

(9) The use of JQ1 does not prove something is a super enhancer, just that it is BRD4 regulated and might be a typical enhancer.

(10) An explanation of how the motifs were identified in E1 is needed. Enrichment over what? Were they purposefully looking for multiple motifs per enhancer? Otherwise what it all comes down to later in the figure is a single motif, and how can that be "enriched"?

(11) A major missing experiment is to deplete rather than over-express SPI1 for the various assays in Figure 4.

(12) The authors start jumping around cell lines, sometimes with little justification. Why is MGC803 used in Figure 4I rather than MKN45? This might be due to more endogenous SPI1. However, this does not make sense in Figure 5M, where ZFP36L is overexpressed in this line rather than MKN45. If SPI1 is already high in MGC803, then the prediction is that ZFP36L1 should already be high. Is this the case?

(13) In Figure 5, HDAC3 should also be depleted to show opposite effects to over-expression (as the latter could be artefactual). Also, direct involvement should be proven by ChIP.

(14) Figure 5G and H are not discussed in the text.

(15) Figure 6C needs explaining. Why are three patients selected here? Are these supposed to be illustrative of the whole cohort? What sub-type of GC are these?

(16) In Figure 6E onwards, they switch to MFC cell line. They provide a rationale but the key regulatory axis should be sown to also be operational in these cells to use this as a model system.

---

## [Referee Report · Reviewer #2 (Public Review)]

Summary:

This manuscript by Wei et al studies the role of ZFP36L1, an RNA-binding protein, in promoting PD-L1 expression in gastric cancer (GC). They used human gastric cancer tissues from six patients and performed H3K27ac CUT&Tag to unbiasedly identify SE specific for the infiltrative type. They identified an SE driving the expression of ZFP36L1 and immune evasion through upregulation of PD-L1. Mechanistically, they show that SPI1 binds to ZFP36L1-SE and ZFP36L1 in turn regulates PD-L1 expression through modulation of the 3'UTR of HDAC3. This mechanism of PD-L1 regulation in gastric cancer is novel, and ZFP36L1 has not been previously implicated in GC progression. However, the data presented are largely correlations and no direct proof is presented that the identified SE regulates ZFP36L1 expression. Furthermore, the effect of ZFP36L1 manipulation elicited a modest effect on PDL1 expression. In fact, several cell lines (XGC1, MNK45) express abundant ZFP36L1 but no PD-L1, suggesting the ZFP36L1 per se is not a key stimulant of PD-L1 expression as IFNg is. Thus, the central conclusions are not supported by the data.

Strengths:

Use of human GC specimens to identify SE regulating PD-L1 expression and immune evasion.

Weaknesses:

Major comments:

(1) The difference in H3K27ac over the ZFP36L1 locus and SE between the expanding and infiltrative GC is marginal (Figure 2G). Although the authors establish that ZFP36L1 is upregulated in GC, particularly in the infiltrative subtype, no direct proof is provided that the identified SE is the source of this observation. CRISPR-Cas9 should be employed to delete the identified SE to prove that it is causatively linked to the expression of ZFP36L1.

(2) In Figure 3C the impact of shZFP36L1 on PD-L1 expression is marginal and it is observed in the context of IFNg stimulation. Moreover, in XGC-1 cell line the shZFP36L1 failed to knock down protein expression thus the small decrease in PD-L1 level is likely independent of ZFP36L1. The same is the case in Figure 3D where forced expression of ZFP36L1 does not upregulate the expression of PDL1 and even in the context of IFNg stimulation the effect is marginal.

(3) In Figure 4, it is unclear why ELF1 and E2F1 that bind ZFP36L1-SE do not upregulate its expression and only SPI1 does. In Figure 4D the impact of SPI overexpression on ZFP36L1 in MKN45 cells is marginal. Likewise, the forced expression of SPI did not upregulate PD-L1 which contradicts the model. Only in the context of IFNg PD-L1 is expressed suggesting that whatever role, if any, ZFP36L1-SPI1 axis plays is secondary.

(4) The data presented in Figure 6 are not convincing. First, there is no difference in the tumor growth (Figure 6E). IHC in Figure 6I for CD8a is misleading. Can the authors provide insets to point CD8a cells? This figure also needs quantification and review from a pathologist.

---

## [Author Response]

**Reviewer #1 (Public Review):**
(1) All the figure legends need to expand significantly, so it is clear what is being presented. All experiments showing data quantification need the numbers of independent biological replicates to be added, plus an indication of what the P-values are associated with the asterisks (and the tests used).

Thank you for your valuable suggestions. We will significantly expand the figure legends to provide a clear and detailed description of the data presented in each figure. Additionally, we will include dot plots in the bar graphs to illustrate the number of independent biological replicates for each experiment. Furthermore, we will specify the statistical tests used for each analysis and include the corresponding P-values associated with the asterisks in the figure legends.

(2) All the Related to point 1, the description of the data in the text needs to expand significantly, so the figure panels are interpretable. Examples are given below but this is not an exhaustive list.

We appreciate your feedback on the clarity of the data description in the text. In response to your suggestion, we will significantly expand the descriptions throughout the manuscript to ensure that each figure panel is fully interpretable. The revised text will provide a more detailed and comprehensive explanation of the data presented.

(3) All the The addition of "super-enhancer-driven" to the title is a distraction. This is the starting point but the finding is portrayed by the last part of the title. Moreover, it is not clear why this is a super enhancer rather than just a typical enhancer as only one seems to be relevant and functional. I suggest avoiding this term after initial characterisations.

Thank you for your thoughtful comment. In this study, the key molecule ZFP36L1 was identified as a target gene through the characterization of the super-enhancer ZFP36L1-SE. The enrichment of H3K27ac at this site meets the threshold defined by the ROSE algorithm, and transcription of ZFP36L1 is regulated by BRD4, making it susceptible to inhibition by the super-enhancer inhibitor JQ1. Although we were unable to directly observe the effects of knocking out the ZFP36L1-SE via Cas9 due to experimental constraints, we believe that the indirect evidence we have gathered is sufficient to demonstrate the super-enhancer's driving role. This approach is consistent with the conventions of previous studies on super-enhancers.

(4) The descriptions of Figures 1B, C, and D are very poor. How for example do you go from nearly 2000 SE peaks to a couple of hundred target genes? What are the other 90% doing? What is the definition of a target gene? This whole start section needs a complete overhaul to make it understandable and this is important as is what leads us to ZFP36L1 in the first place.

We appreciate your feedback and apologize for the confusion caused by the initial descriptions. As described in the manuscript, the function of SE peaks depends on their location. Figure 1C shows the distribution of these peaks, where "Over 50% of these peaks were located in the non-coding regions such as exons and introns, and their predicted target genes were transcribed to produce non-coding RNAs; the peaks distributed in transcription start and termination sites activated the promoters and directly drove the transcription of protein-coding genes". Our research focuses on protein-coding genes, and we apologize for any misunderstanding due to the inadequate description. We will provide additional clarification to make this distinction clear.

(5) It is impossible to work out what Figures 1F, H, and I are from the accompanying text. The same applies to supplementary Figure S1D. Figure 1G is not described in the results.

Thank you for pointing out these issues. We will make the necessary revisions to provide additional explanations for Figures 1F, H, I, G, and supplementary Figure S1D.

(6) What is Figure 2A? There is no axis label or description.

Thank you for bringing this to our attention. We will add the missing axis labels and provide a detailed description for Figure 2A to ensure clarity and accurate interpretation.

(7) Why is CD274 discussed in the text from Figure 2E but none of the other genes? The rationale needs expanding.

CD274 (also known as PD-L1) is a key focus of our subsequent research. The other immune checkpoints are not expressed on tumor cells but rather on immune cells. We will provide additional explanation in the text to clarify this distinction.

(8) Figure 2G needs zooming in more over the putative SE region and the two enhancers labelling. This looks very strange at the moment and does not show typical peak shapes for histone acetylation at enhancers.

We appreciate your feedback. Our intention with Figure 2G was to present the position of ZFP36L1-SE at a macro level rather than focusing on specific details. This broader view is meant to provide context for the SE region in relation to the surrounding genomic landscape.

(9) The use of JQ1 does not prove something is a super enhancer, just that it is BRD4 regulated and might be a typical enhancer.

Thank you for your comment. The role of JQ1 as a super-enhancer inhibitor has been widely reported and recognized in the literature. Its use in experimental studies targeting super-enhancers is a well-established practice. We acknowledge that while JQ1 inhibition indicates BRD4 regulation, it is consistent with the identification of super-enhancers as well.

(10) An explanation of how the motifs were identified in E1 is needed. Enrichment over what? Were they purposefully looking for multiple motifs per enhancer? Otherwise what it all comes down to later in the figure is a single motif, and how can that be "enriched"?

Thank you for your feedback. We used the MEME-ChIP online tool for motif identification, which is a widely recognized method in transcription factor research. MEME-ChIP applies established algorithms to identify known motifs within DNA sequences. For detailed information on the tool's working principles and algorithms, please refer to the reference provided and the URL included in the Materials and Methods section of our manuscript. MEME-ChIP: https://meme-suite.org/meme/tools/meme.

(11) A major missing experiment is to deplete rather than over-express SPI1 for the various assays in Figure 4.

We apologize for this oversight and acknowledge that the depletion of SPI1, in addition to over-expression, would have provided a more comprehensive analysis. Due to experimental constraints, we are unable to include this depletion experiment in the current study. We appreciate your understanding and will consider this suggestion for future research.

(12) The authors start jumping around cell lines, sometimes with little justification. Why is MGC803 used in Figure 4I rather than MKN45? This might be due to more endogenous SPI1. However, this does not make sense in Figure 5M, where ZFP36L is overexpressed in this line rather than MKN45. If SPI1 is already high in MGC803, then the prediction is that ZFP36L1 should already be high. Is this the case?

Thank you for your feedback. We want to clarify that we are not arbitrarily jumping between cell lines. Each experiment was validated in two different cell lines. We aimed to present representative results within the constraints of the manuscript, but if more detailed results from additional cell lines are needed, we can provide them upon request. Regarding your concern, results from the MKN45 cell line are consistent with those observed in MGC803, and these findings are not influenced by SPI1 or ZFP36L1 expression levels.

(13) In Figure 5, HDAC3 should also be depleted to show opposite effects to over-expression (as the latter could be artefactual). Also, direct involvement should be proven by ChIP.

We appreciate your feedback. We acknowledge that depleting HDAC3, in addition to overexpressing it, would provide a more comprehensive analysis. Unfortunately, due to experimental constraints, we are unable to include this depletion experiment in the current study. We recognize these limitations and appreciate your understanding. We will consider these aspects for future research. Additionally, we would like to clarify that HDAC3 is a histone deacetylase and not a transcription factor, so it does not directly bind to DNA and therefore is not suitable for ChIP analysis.

(14) Figure 5G and H are not discussed in the text.

Thank you for pointing this out. We will include a discussion of Figures 5G and H in the revised manuscript. The additional details should provide the necessary context and interpretation for these figures.

(15) Figure 6C needs explaining. Why are three patients selected here? Are these supposed to be illustrative of the whole cohort? What sub-type of GC are these?

Thank you for your comment. The three patients with infiltrative GC shown in Figure 6C were selected as representative images based on prior reviewer suggestions.

(16) Figure 6E onwards, they switch to MFC cell line. They provide a rationale but the key regulatory axis should be sown to also be operational in these cells to use this as a model system.

Thank you for your comment. We would like to clarify that we used the MC38 cell line, which is a colon cancer cell line, rather than MFC. Our focus was on demonstrating the role of ZFP36L1 in vivo, rather than specifically discussing the regulatory axis in this context. We chose MC38 cells instead of MFC cells due to practical considerations. Specifically, MFC cells were shown in our experiments to be unable to form tumors in wild-type mice, despite previous reports suggesting their tumorigenicity. We will provide a rationale for this choice in the manuscript. We acknowledge that validating the entire regulatory axis in the MC38 cell line would enhance the study's depth. However, due to experimental constraints, we are unable to complete this additional validation. We appreciate your understanding and will consider this aspect for future research.

**Reviewer #2(Public Review):**
(17) The difference in H3K27ac over the ZFP36L1 locus and SE between the expanding and infiltrative GC is marginal (Figure 2G). Although the authors establish that ZFP36L1 is upregulated in GC, particularly in the infiltrative subtype, no direct proof is provided that the identified SE is the source of this observation. CRISPR-Cas9 should be employed to delete the identified SE to prove that it is causatively linked to the expression of ZFP36L1.

Thank you for your thoughtful comment. In this study, the key molecule ZFP36L1 was identified as a target gene through the characterization of the super-enhancer ZFP36L1-SE. The enrichment of H3K27ac at this site meets the threshold defined by the ROSE algorithm, and transcription of ZFP36L1 is regulated by BRD4, making it susceptible to inhibition by the super-enhancer inhibitor JQ1. Although we were unable to directly observe the effects of knocking out the ZFP36L1-SE via Cas9 due to experimental constraints, we believe that the indirect evidence we have gathered is sufficient to demonstrate the super-enhancer's driving role. This approach is consistent with the conventions of previous studies on super-enhancers.

(18) In Figure 3C the impact of shZFP36L1 on PD-L1 expression is marginal and it is observed in the context of IFNg stimulation. Moreover, in XGC-1 cell line the shZFP36L1 failed to knock down protein expression thus the small decrease in PD-L1 level is likely independent of ZFP36L1. The same is the case in Figure 3D where forced expression of ZFP36L1 does not upregulate the expression of PDL1 and even in the context of IFNg stimulation the effect is marginal.

Thank you for your detailed observations. In our study, the regulatory effect of ZFP36L1 on PD-L1 was validated at the mRNA level, protein level, and through flow cytometry, with each experiment being repeated multiple times. The results of the Western blot were quantitatively assessed using densitometry rather than relying solely on visual inspection. It is important to note that interferon-gamma (IFNγ) stimulation significantly enhances PD-L1 expression, which under the same exposure conditions, may make the baseline expression of PD-L1 appear unchanged. This could explain the marginal effect observed under IFNγ stimulation.

(19) In Figure 4, it is unclear why ELF1 and E2F1 that bind ZFP36L1-SE do not upregulate its expression and only SPI1 does. In Figure 4D the impact of SPI overexpression on ZFP36L1 in MKN45 cells is marginal. Likewise, the forced expression of SPI did not upregulate PD-L1 which contradicts the model. Only in the context of IFNg PD-L1 is expressed suggesting that whatever role, if any, ZFP36L1-SPI1 axis plays is secondary.

Thank you for your insightful comments. First, ELF1, E2F1, and SPI1 were predicted transcription factors, and experimental validation is crucial. Our results specifically demonstrate that only SPI1 binds to ZFP36L1-SE, while ELF1 and E2F1 do not, confirming the specificity of SPI1. Second, Second, as mentioned in point (18), experimental results, such as those from western blot, should not be evaluated by eye alone. Our findings are quantitatively assessed, and the regulatory relationships have been confirmed through repeated experiments. This finding is supported by multiple experimental validations, including mRNA, protein, and flow cytometry analyses. Furthermore, using IFNγ to study the regulation of PD-L1 is a common and widely accepted approach in this field. Many studies adopt this model, and it should not be concluded that the axis is secondary simply because PD-L1 expression is observed primarily under IFNγ stimulation. Similarly, other popular research areas, such as ferroptosis and autophagy, also use specific inducers, but this does not diminish the significance of the pathways being studied.

(20) The data presented in Figure 6 are not convincing. First, there is no difference in the tumor growth (Figure 6E). IHC in Figure 6I for CD8a is misleading. Can the authors provide insets to point CD8a cells? This figure also needs quantification and review from a pathologist.

Regarding this observation, we will provide an explanation in the discussion section: "Several studies have proposed that reducing PD-L1 expression enhances the tumor-killing effect of cytotoxic T lymphocytes in vitro and reduces primary tumor foci in vivo. Conversely, findings from this study suggest that PD-L1 expression is associated with immune evasion in metastatic foci." We are unsure why those studies concluded that PD-L1 expression levels would impact the size of the primary tumor. We are more inclined to support the perspective of John et al.Klement JD, Redd PS, Lu C, et al. Tumor PD-L1 engages myeloid PD-1 to suppress type I interferon to impair cytotoxic T lymphocyte recruitment. Cancer Cell. 2023;41(3):620-636.e9. doi:10.1016/j.ccell.2023.02.005

**Reviewer #1 (Recommendations For The Authors):**
(21) Supplementary Figure 1 lacks a legend.

We will add the legend for Supplementary Figure 1.

(22) Figure 1E, data from "expanding" GC samples is not discussed.

We will add a discussion of the "expanding" GC samples in the manuscript.

(23) How are "high" and "low" defined in Figure 2A, right?

Thank you for your question. In Figure 2A, the "high" and "low" categories on the x-axis are derived from the Friends analysis. This analysis is designed to compare the similarity between different genes or gene sets based on semantic similarity metrics from Gene Ontology (GO). The x-axis represents the semantic similarity score, which reflects how closely related the functions of the genes or gene sets are. This helps in identifying the most significant genes or those related to specific pathways or cell types of interest.

GOSemSim[2.22.0]

Yu G, Li F, Qin Y, Bo X, Wu Y, Wang S. GOSemSim: an R package for measuring semantic similarity among GO terms and gene products. Bioinformatics. 2010;26(7):976-978. doi:10.1093/bioinformatics/btq064

(24) Font sizes in multiple figures need to increase. For example, Figure 2C (but many other places).

The font sizes in the figures, including Figure 2C, will be increased as requested.

(25) Figure 4K assays TE activity, not SE as stated in the text.

SEs are composed of multiple TEs. ZFP36L1-E1 is a core element of the ZFP36L1-SE. Due to the excessive length of the ZFP36L1-SE sequence, it was not feasible to insert the entire SE into a dual-luciferase reporter plasmid. It is a common practice to validate such experiments by inserting the typical enhancer elements instead.

(26) In Figure 6I, why is CD8 shown? What is the reason for choosing this?

CD8α is primarily used to assess immune evasion by tumor cells against T-cell cytotoxicity. CD8α is typically negatively correlated with PD-L1 expression and serves as an indicator of T-cell infiltration.

(27) The discussion should be more focussed. The majority of this is general stuff about either super enhancers or PD-L1 regulation. This should be curtailed and more pertinent things retained.

We will revise the discussion to be more focused. The content will be streamlined to emphasize the most pertinent points related to our study.

**Reviewer #2 (Recommendations For The Authors):**
(28) In Figure 1H various immune cell populations differ between the two types of GC. Unclear what is the biological significance in the context of ZFP36L1.

The results in Figure 1H provide insight into the SE-driven immune escape signatures of infiltrative gastric cancer (GC). These findings help to contextualize the role of ZFP36L1 in modulating the tumor microenvironment, particularly in relation to immune cell infiltration and immune evasion mechanisms.

(29) A bivalent profile for H3K27ac is also observed in expanding gastric cancer (Figure 1B), not only in infiltrating GC as the authors claim.

We did not intend to imply that bivalent H3K27ac enrichment is exclusive to infiltrating gastric cancer. In fact, super-enhancers were identified in both expanding and infiltrative GC. Our point was to highlight that the bivalent enrichment profile is more pronounced in infiltrative GC.

(30) There is a typo in line 81.

The typo in line 81 will be corrected. Thank you for pointing it out.